# Focus on Innovation or Focus on Sales? The Influences of the Government of China's Demand-Side Reform during COVID-19 and Their Sustainability Consequences in the Consumer Products Industry

**Guangying Xie [1], Shengyan Wu [2],\* and Zhengjiang Song [2]**

1    School of Economics and Management, China University of Mining and Technology, Xuzhou 221116, China
2    School of Medicine and Health Management, Guizhou Medical University, Guiyang 550031, China
\*    Correspondence: wushengyan@gmc.edu.cn

**Abstract:** Affected by COVID-19, the Chinese government has implemented demand-side reform policies to stimulate consumption. In this context, should consumer goods companies focus on innovation (FOI) or focus on ales (FOS), and what impact does it have on sustainability? Based on the empirical data of listed companies in China's A-share consumer goods industry, this paper uses a hierarchical regression model, a mediation effect test, moderation effect analysis, and a robustness test, and it finds that for consumer products industry enterprises under the background of demand-side reform: (1) FOI negatively affects enterprise sustainability, FOS positively affects enterprise sustainability, and tilting resources from FOI to FOS is conducive to improving sustainability; (2) the direct effect of FOI on sustainability is not significant, and its negative effect on sustainability is exerted exclusively indirectly through FOS; and (3) firm value can positively moderate the relationship between FOS and sustainability, but its moderating effect on the relationship between FOI and sustainability is insignificant. These conclusions are of positive significance for the study of corporate innovation, sales behavior, and sustainability performance under demand-side reform. More broadly, this paper enriches the study of corporate sustainability in the context of an unfavorable macro environment and short-term and large policy stimulus in the market.

**Keywords:** sustainability; innovation; sales; firm value; demand-side reform; market opportunity; ESG; focus on innovation; focus on sales

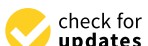

## 1. Introduction

Sustainability is a crucial strategic objective for many firms nowadays, particularly under the background of serious threats of environmental problems [1–3]. However, extant research indicates that the linkage between sustainability and financial performance is not always significant, no matter in the short-term or in the long-term, which causes insufficient incentives of sustainability through innovation and other pathways in many firms [4–6]. Furthermore, industrial differences also exist, especially between heavy polluting industries and the consumer products industry [7,8]. In highly polluting industries, governments often require companies to increase innovation and take various measures to reduce pollution and meet required national, industrial, or other standards. This means that the sustainability of highly polluting industries is not entirely voluntary but is forced to a certain extent.

On the contrary, there is no rigorous law requirement in regards to innovation and sustainability by the government in the consumer products industry due to its low pollution nature. Therefore, firms' sustainable behavior is more voluntary, and they have more discretion in deciding whether to increase investment in innovation and sustainability or not. In this circumstance, financial performance becomes the major motivation, and

there are mainly two approaches for firms in the consumer products industry to strengthen sustainability. One is to focus on innovation (FOI), including technological innovation, process innovation, management innovation, etc., which may raise efficiency and reduce cost [9,10]. The other is to focus on sales (FOS), particularly, to strength the network between the sales force and customers, which may increase market-oriented communication and customer value [11].

Facing global and civil economic recession brought by COVID-19, China's household consumption suffered greatly in 2020. Many small and medium firms of the consumer products industry encountered poor sales and survival dilemmas. The government implemented a policy called demand-side reform at the end of 2020 to help firms to pull through by adopting multitudinous measures covering political, economic, financial, social, etc., dimensions. One of the most attractive measures for firms was the consumption coupon provided directly to consumers by the government, which cut the actual payment price down from the consumer side through public financial expenditures. To seize upon this market opportunity, should companies focus on innovation or sales, and what impacts do they have on sustainability? This is the main question of this study. Moreover, considering the significance of FOI and its probable impact on FOS, this paper further studies its direct effects on sustainability and indirect effects through FOS. In addition, firm value is introduced as a moderator to explore its influence on the relationship between FOI, FOS, and sustainability.

Motivated by the relationship between FOI, FOS, and their impacts on sustainability in the consumer products industry under the background of China's demand-side reform, this paper applied public data to empirically address the above questions. To be specific, we first conducted a correlation analysis and hierarchical regression model to observe the influence of FOI and FOS on sustainability. Results manifested that FOI negatively affects sustainability, while FOS's impact is positive. Then, the mediation effect test of FOS between FOI and sustainability was examined, which indicated that FOS fully mediates the adverse effect of FOI on sustainability. To go further, we explored and found, through the moderating effect analysis method, that firm value positively moderates the relationship between FOS and sustainability. Nevertheless, firm value's moderating effect between FOI and sustainability is insignificant. Two kinds of robustness tests were conducted to make the results more convincing.

This paper contributes to the current literature in several ways. First, we found that for the consumer goods industry, when short-term and huge promotional opportunities arise in the market environment, FOI has a negative impact on the sustainability of the enterprise, while FOS can promote the sustainable improvement of the enterprise. At this point, FOS is more beneficial than FOI to the sustainability of the business. This new context and conclusion challenges traditional research to emphasize that companies should focus on innovation rather than sales [9,10]. Second, further research shows that the negative impact of FOI on sustainability is entirely mediated by FOS. This result indicates that FOI itself does not compromise sustainability, and the main reason for its negative impact on sustainability in the context of demand-side reform is to crowd out the resources needed by FOS, which has a significant advantage in seizing market opportunities to improve sustainability. This finding has implications for differentiating industries and contexts to study the impact of innovation on sales and sustainability [5,8]. Third, this paper further explores the role of firm value in the relationship between FOI, FOS, and corporate sustainability and finds that in demand-side reform, large enterprises are more able to seize opportunities and improve corporate sustainability through FOS than small enterprises, but there is no significant difference between the two in terms of FOI and sustainability. This result enriches research on the impact of firm value on innovation, sales, and sustainability and opens new explorations in the context of demand-side reform [12].

The rest of this paper is organized as follows: Section 2 presents a brief literature review related to innovation, sales, and sustainability and proposes corresponding hypotheses. Section 3 illustrates the data collection process and research methodology. Section 4

shows the statistical results and discusses the findings. Section 5 explores the insights and concludes the paper with a discussion on research limitations and future research directions.

## 2. Literature Review and Hypotheses Development

### 2.1. The Concept of Sustainability and its Antecedents

Due to the increasing number of environmental, social, and governmental challenges worldwide, the concept of sustainability has been at the center of today's environment and living surroundings debate since the definition of sustainable development given by the World Commission on Environment and Development in 1987, which proposed that sustainable development is a kind of development that meets the needs of current generations without compromising the ability of future generations to meet their needs and aspirations [13,14]. In accordance with this definition, it is widely acknowledged that social dimension, environmental dimension, and economic dimension are the three dominating pillars of sustainability [15,16]. For firms, corporate sustainability is connected to the extent to which firms and their products and service quality meet the environmental, social, and governance dimensions in the view of organizational perspectives, and it has been linked to the effects that firms can provide to society when playing a state role and substituting the functions of governments from the macro perspective [17–19].

Previous studies have revealed a wide range of business benefits that accrue to firms engaged in sustainability, for instance, efficiency, new product development, sales growth, customer satisfaction, firm performance, firm image, and reputation [20–22]. Several major antecedents including regulations, institutional environments, firm reputations, innovation strategies, technological advancements, customer demand, and competitive pressure are examined [23–25]. Among them, FOI and FOS are two crucial internal strategies that firms may deploy in the context of demand-side reform policy.

In this study, we defined FOI as a survival philosophy that is mainly oriented by innovation, strategic decision-making, business activities, and resource allocation of enterprises centered on innovation, and the purpose of an enterprise's operation is to strive to improve productivity and provide customers with innovative and high value-added products and services. Through FOI, companies can increase production efficiency, optimize the quality of products and services, reduce pollution and resource consumption, better meet the needs of various stakeholders from the production side, and improve the sustainability of enterprises [9,10].

We defined FOS as a sales-oriented survival philosophy, in which the company's strategic decision-making, business activities, and resource allocation are based on sales and marketing, and the purpose of an enterprise operation is to expand the market scope, improve market share and influence, and establish good stakeholders and social public relations, so as to succeed in the market. By FOS, enterprises may expand the distribution channels and market influence, improve the supplier–distributor relationship, directly provide price discounts and various benefits to consumers, better meet the needs of various stakeholders from the market side, and enhance the sustainability of the enterprise [11].

### 2.2. The Impacts of FOI and FOS on Sustainability

Generally, innovation, which includes organizational innovation, market innovation, process innovation, product innovation, and service innovation, can improve corporate sustainability on many fronts, such as customer demand, financial performance, competitive advantage, and so on [26–28]. Moreover, innovation may also target beyond economic goals by aiming at social and environmental purposes and thus trigger businesses towards sustainability priorities [29,30]. Nevertheless, there is also evidence indicating that the innovations and technologies already visible and those in the pipeline clearly may lead to corporate power, social inequality, and certain controversial effects on the environment and high-risk activities in the future, some of which may even be global and systemic [31].

What is more, firms need to respond to the environment effectively and efficiently when the environment is uncertain and when consumers are more price sensitive in

emerging countries [10]. However, sustainable policies often require special investments in technology, innovation, products, etc., resulting in increased costs [32,33]. Consequently, there are still many companies that view sustainability as a legal and social obligation requiring investments that may never be recovered rather than an opportunity [34]. Once they meet the regulatory standards set by the government or the sustainability standards required by the market, they will reduce their investment in innovation [35]. Based on this view, sustainability should be achieved in an efficient way. Although innovation may improve sustainability, the characteristics of being long-term, uncertain, and requiring constant commitment of large capital expenditures determine that it is not fast and highly-efficient [36]. In the context of demand-side reform, the consumer products industry is facing short-term and massive market opportunities. From the perspective of sustainability and short-term performance, if enterprises still insist on investing their main capital in innovation, it may waste corporate resources and resilience to a certain extent, which is not conducive to quickly seizing market opportunities and improving current sustainability. Thus, we propose the following:

**Hypothesis 1.** *In the context of demand-side reform, FOI has a negative impact on sustainability.*

In recent years, with the concerns of consumers with product quality, safety, environmental protection, and other requirements, sustainability issues are receiving more and more attention in the consumer goods industry [37,38]. While research has given substantial attention to drivers of consumption and market share of existing sustainable products and sustainability's impact on sales, studies have given limited attention to the influence of FOS on firm sustainability [39,40]. Improving sustainability can not only be achieved through traditional technological innovation functions, but it can also be obtained through human capital, social networks, and other means, which are associated with sales and the market [41,42]. Research findings indicate that focusing on sales, by allocating resources to marketing departments, building good market and stakeholder relationships and networks, and strengthening interaction can also promote corporate sustainability [38,43]. Particularly, in the context of demand-side reform, in order to seize market opportunities, building good market relations is more conducive to the growth of corporate performance and short-term sustainable development. Hence, we propose the following:

**Hypothesis 2.** *In the context of demand-side reform, FOS has a positive impact on sustainability.*

### 2.3. The Influence of FOI on FOS

Innovation and sales are two important functions of the enterprise. The main purpose of enterprise innovation is to improve the market competitiveness of products and services and obtain a favorable market position and corporate performance [44–46] One of the key avenues is to boost sales. Previous studies have shown that corporate innovation has a significant boosting effect on sales and customer relationships by providing better products and services [47–49]. In fact, innovation and marketing are mutually reinforcing. Innovative products and services are conducive to enterprises opening up the market and expanding market share. Good marketing relationships and sales capabilities in turn promote innovation and R&D activities. For example, companies can obtain market changes and the latest needs of customers through sales, thus providing direction for innovation [50–52]. Therefore, in general, in the normal business process of the enterprise, innovation is conducive to the expansion of the market, increasing sales, meeting customer needs, and improving customer relations. Through FOI strategies, companies can promote sales to a certain extent.

However, on the other hand, both innovation and sales require the consumption of resources of the enterprise. Although they can promote each other to some extent, there will be competition between them in the case of limited or scarce resources [53–55]. For enterprises, FOI means concentrating resources and capabilities on innovation, which

will inevitably affect the investment of resources and capabilities in sales. Conversely, if companies choose FOS, it will inevitably affect the investment of resources and capabilities in innovation. In the context of demand-side reform, the originally sluggish market suddenly appears to have many opportunities in the short term. Companies that want to seize upon these opportunities need to rapidly increase their resource investment in sales and marketing. However, even for resource-rich enterprises, it is not easy to quickly raise so many resources in such a short period of time, which will cause resource shortages or temporary resource misalignment within the enterprise [56]. At this time, if the enterprise still adheres to innovation and invests resources around innovation, it will weaken the resource investment ability of enterprises in sales and marketing to a certain extent. Hence, we propose the following:

**Hypothesis 3.** *In the context of demand-side reform, FOI has a positive impact on FOS.*

### 2.4. The Moderating Effects of Firm Value

In the relationship between innovation, sales, and sustainability, firm value is an important variable factor. As a major measure of business size, firm value reflects the level of resources and capabilities a business has. Generally, large enterprises with higher market value would have more resources and capabilities, and therefore the level of innovation may be higher [57–59]. Furthermore, large corporations would receive more attention from society and the public, and they are more pressured and motivated to perform better in terms of sustainability [60,61]. However, in the specific context of this study, the impact of FOI on corporate sustainability may be negative based on the analysis above. Under the background of demand-side reform, the negative impact of innovation on sustainability is mainly due to its slow response to changes in the market environment and resource occupation [12,62]. Demand-side reform requires companies to quickly seize the opportunities that suddenly appear in the market, while with innovation, is it difficult to achieve a rapid response in a short period of time because it is a relatively slow process from input to output. At this time, if resources and capabilities are mainly concentrated on innovation, it will cause a certain degree of waste of resources. For large enterprises, they have more advantages than small enterprises in terms of market influence and public relations [63]. If they are too focused on innovation, there may be a more adverse impact on the sustainability of enterprises. Hence, we propose the following:

**Hypothesis 4.** *In the context of demand-side reform, firm value negatively moderates the relationship between FOI and sustainability.*

Firm value may not only affect the relationship between FOI and corporate sustainability but also the relationship between FOS and corporate sustainability. Although we argue in Hypothesis 2 that FOS has a significant positive impact on corporate sustainability, this impact may still vary significantly between businesses of different sizes. Large enterprises have more competitive advantages in relying on FOS to improve their enterprise capabilities due to their advantages in resource endowments and market capabilities. Specifically, the impact of corporate value on FOS and corporate sustainability is mainly twofold. On the one hand, large enterprises have advantages over small- and medium-sized enterprises in terms of product quality, safety and environmental protection, customer service, profitability, etc.; if more resources are invested in sales and market competition, it is easier to receive the welcome and recognition of the market and customers and to obtain a higher social reputation and corporate sustainability [64–66]. On the other hand, large enterprises often have good social network relationships with governments, suppliers, and other key stakeholders compared with small and medium firms [32,67,68], which is more able to improve the sustainability of enterprises by increasing sales resource inputs and expanding sales in the face of government-led demand-side reform opportunities. Hence, we propose the following:

**Hypothesis 5.** *In the context of demand-side reform, firm value positively moderates the relationship between FOS and sustainability.*

*2.5. Conceptual Model*

Affected by COVID-19, the growth rate of residents' incomes has slowed down, and the consumption momentum is insufficient. For the consumer goods industry, the impact has been particularly severe, and some small- and medium-sized enterprises have even hovered on the brink of survival. In this context, while most companies and their executives know that innovation is beneficial to their long-term sustainability, they face short-term survival dilemmas. For enterprises, the pressure to survive is heavier. At this time, the demand-side reform led by the central and local governments have stimulated the consumption demand of residents to a large extent by issuing consumption coupons and consumption subsidies to consumers, and it also created favorable conditions for the recovery of the short-term performance of enterprises.

Previous research has shown that corporate innovation is not only good for sustainability, but it also promotes sales and the maintenance of good stakeholder relationships [11,28,48]. However, the influencing process of innovation is often indirect and slow [44]. In contrast, short-term performance in seizing market opportunities is better in terms of direct-to-market promotions, sales promotion, and public relations. Against the backdrop of the pressure of COVID-19 and the market opportunities brought about by demand-side reform, the innovation and sales decisions of enterprises have become issues worth exploring. In this paper, the FOI and FOS decisions of enterprises and their impacts on corporate sustainability are first analyzed, and the research assumptions from H1 to H3 are proposed. Then, in order to explore the differential performance of enterprises of different sizes between FOI, FOS, and sustainability, we continue to analyze the moderating effect of firm value on the relationship between FOI and sustainability and FOS and sustainability and propose research hypotheses H4 and H5. Based on the above analysis and assumptions H1 to H5, we preliminarily construct the theoretical model of this paper, as shown in Figure 1.

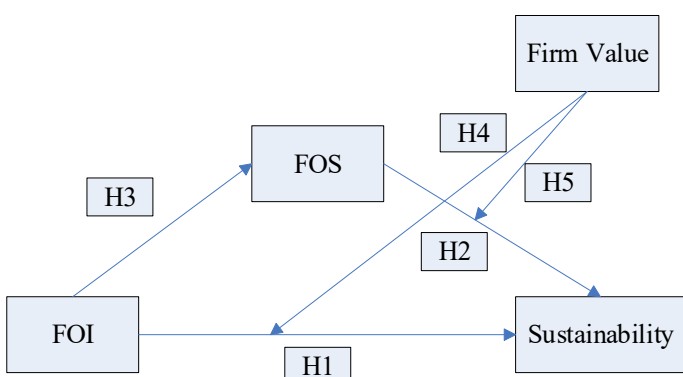

**Figure 1.** The conceptual model of this study.

## 3. Data and Methodology

*3.1. Samples and Data Collection*

The data samples of this study are mainly from the consumer products industry enterprises listed on China's A-share market, including food and main supplies in retail, food, beverages and tobacco, household and personal products, automobiles and auto parts, durable consumer goods and clothing, consumer services, media, and other industries, which comprehensively cover the daily consumption and optional consumption in the Wind company's industry classification system, which can better reflect the overall situation of China's consumer market. Wind is a comprehensive database company that collects data from Listed Companies in China, and its corporate financial data and ESG data are widely used in academic research [69–71]. Considering the authority and verifiability of the data

sources, we used the Wind database's second-hand data in this study. As of 31 August 2021, the total number of A-share listed consumer products industry enterprises on the Shanghai Stock Exchange and the Shenzhen Stock Exchange was 920. Among them, 885 firms were included in the ESG (Environmental, Social, and Governance) rating of the Sino-Securities Index, 138 firms were included in the Wind ESG rating, and 123 firms were included in the FTSE (Financial Times Stock Exchange) Russell ESG rating list. Considering the comprehensiveness of the data selection, this study used 885 consumer products industry enterprises included in the ESG rating list of Sino-Securities Index as the research object, and after eliminating the missing values, 723 sample companies were selected (two firms that only had missing value in firm value were kept), and the specific data came from the financial report and the Sino-Securities ESG Index of listed companies in the Wind database.

*3.2. Variables*

3.2.1. Dependent Variable

The core dependent variable of this study is sustainability, the current academic measurement methods of which can be divided into two categories: one is to directly use the data and evaluation indicators of sustainability rating agencies, such as the domestic Sino-Securities ESG rating, the FTSE Russell rating, the Wind ESG rating, etc., whose advantages are more objective and fair [68,72]. The data obtained by professional research institutions in sustainability management practices and evaluations are more reflective of objective reality, but there are deficiencies in the incomplete data of some small- and medium-sized enterprises, which cannot reflect the specific dimensions or aspects of sustainability management required by researchers. The other type is to design empirical indicators by researchers themselves, and measure and reflect upon one or more dimensions of sustainability [26,36,73]. The advantage is that the data access is convenient, objective, and fair, and it can cover the vast majority of the listed company samples, while the disadvantage is that it does not fully reflect the various dimensions of corporate sustainability performance.

Considering sample coverage and the conceptual composition of environmental performance, this study uses the sustainability ranking data compiled by the Shanghai Sino-Securities Index Information Service Co., Ltd., for two main reasons: First, in terms of sample coverage, among the 920 companies in the consumer products industry in the A-share market of China, the Sino-Securities Index covers 885 companies, with a coverage rate of 96.20%, which can better reflect the overall status of the sample; second, in terms of index composition and data reliability, the ESG score of the Sino-Securities Index comprehensively considers the internal management system, business objectives, green products, external certification, violations, and other factors, and the basic data used are mainly based on the public disclosure data of listed companies and integrate auxiliary information such as social responsibility reports, sustainable development reports, national regulatory announcements, news media data, etc., with strong credibility. Specifically, considering the time lag effect of demand-side reform on the consumer products industry, this paper used the ESG score rating data on 30 June 2021 (the actual rating time was 30 April 2021) and converted the nine grades of C/CC/CCC/B/B/BB/BBB/A/AA/AAA in the sustainability performance rating into numerical variables from 1 to 9, respectively, to simplify the quantitative analysis. To be specific, C has the lowest grade and was assigned the value of 1. AAA has the highest rating and was assigned the value of 9. Among the other grades, the CC grade was assigned the value of 2, the CCC grade was assigned the value of 3, the B grade was assigned the value of 4, the BB grade was assigned the value of 5, the BBB grade was assigned the value of 6, the A grade was assigned the value of 7, and the AA grade was assigned the value of 8.

3.2.2. Predictor and Moderator Variables

There were two main explanatory variables in this study: FOI and FOS. FOI is an important explanatory variable in this study, reflecting the resource investment and propensity of enterprises in R&D and innovation. Most of the empirical studies draw on the

practices of Miller and Le Breton-Miller [74] and measure the ratio of annual R&D expenses to annual sales. Using this approach, we measured FOI using the ratio of R&D expenses to sales revenue in the third quarter of 2021. FOS is another key explanatory variable of this research, which is a part of marketing orientation and reflects the investment and propensity of enterprises in personnel sales and sales relationship network construction. The proportion of sales personnel was used as a measurement variable of FOS in this study. Specifically, in order to better study the impact of FOS under demand-side reform, the data of the previous day proposed by the national demand-side reform policy, that is, the specific data of 10 December 2020, were used.

The moderator variable in this study was enterprise value, which refers to the overall value of all assets of the enterprise. It is calculated by adding up the fair market value of equity and the fair market value of debt. In practice, the carrying amount of interest-bearing debt is usually substituted for the fair market value of a debt. Specifically, this study used China's Wind database's statistical method:

Enterprise value = equity value + debt value;

Equity value = China A share market closing price * China A share market total + China B share closing price * China B share market total * RMB foreign exchange price + China H share closing price * China H share market total * RMB foreign exchange quote price + overseas listed stock closing price * overseas listed stock total * RMB foreign exchange quote price.

Debt Value = Total Liabilities − Interest-free Current Liabilities − Interest-Free Non-Current Liabilities.

### 3.2.3. Controls

Considering that factors such as return on assets, enterprise risk, employee size, and executive characteristics may have significant impacts on the FOS and sustainability performance of enterprises [58,75], some important related variables were also controlled in the research process, mainly including return on total assets, net asset liability ratio, total number of employees, proportion of executives, and proportion of independent directors. The data of the third quarter of 2020 were used for the implementation date of the demand-side reform.

### 3.3. Research Methodology

Focusing on the relationship between FOI, FOS, and sustainability in the consumer products industry under demand-side reform, and exploring the moderating role of enterprise value, this paper proposed to use classic methods including general linear regression, the mediation effect test [76], the moderation effect test [77], and the robustness test [78], and it divided the empirical research into four parts: (1) test the impacts of FOI and FOS on the sustainability of enterprises; (2) test the impacts of FOI on the sustainability of the enterprise through FOS; (3) test the moderating effect of enterprise value on FOI and FOS to sustainability; (4) carry out robustness tests by estimating with the robust regression method and replacing environmental ratings to ESG scores.

## 4. Results

### 4.1. Descriptive Statistics and Correlation Analysis

Before performing regression analysis to test the research hypotheses, we calculated the descriptive statistics and correlation coefficients in Table 1. There were 723 samples for all the variables except firm value, the valid sample quantity of which was 721. The mean value of sustainability was 6.32, with a standard deviation of 1.41, suggesting significant variation in the performance of sustainability. The average for FOI was 3.04, with a minimum value of 0 and a maximum value of 25.63, which indicates that there was a huge difference in innovation investment between consumer products industry enterprises. Strong variation also existed in FOS, the standard deviation of which was 20.41, larger than its mean value. The Pearson correlations revealed that sustainability had negative relationships with FOI,

while its coefficient of correlation with FOS was a positive number. Firm value, return on assets, debt to net worth ratio, total employees, and executive proportion were all significantly associated with sustainability.

**Table 1.** Descriptive and correlation statistics of variables.

| Variables | 1 | 2 | 3 | 4 | 5 | 6 | 7 | 8 | 9 |
|---|---|---|---|---|---|---|---|---|---|
| 1. Sustainability | 1.000 | | | | | | | | |
| 2. FOI | −0.151 *** | 1.000 | | | | | | | |
| 3. FOS | 0.169 *** | −0.262 *** | 1.000 | | | | | | |
| 4. Firm value (Million RMB) | 0.129 *** | −0.079 ** | −0.029 | 1.000 | | | | | |
| 5. Return on assets | 0.222 *** | −0.136 *** | 0.021 | 0.222 *** | 1.000 | | | | |
| 6. Debt to net worth ratio | −0.092 ** | −0.057 | −0.017 | −0.008 | −0.027 | 1.000 | | | |
| 7. Total employees | 0.226 *** | −0.084 ** | −0.011 | 0.428 *** | 0.042 | 0.018 | 1.000 | | |
| 8. Executives proportion | −0.216 *** | 0.163 *** | −0.092 ** | −0.104 *** | −0.016 | −0.049 | −0.219 *** | 1.000 | |
| 9. Ratio of independent directors | −0.059 | 0.015 | −0.023 | 0.096 *** | 0.043 | −0.026 | 0.069 * | 0.059 | 1.000 |
| Mean | 6.317 | 3.039 | 19.208 | 24072.579 | 4.997 | 1.273 | 6417.055 | 0.004 | 0.380 |
| Minimum | 1.000 | 0.000 | 0.260 | 0.418 | −18.889 | −22.048 | 59 | 0.000 | 0.300 |
| Maximum | 9.000 | 25.627 | 91.460 | 2312911.830 | 52.264 | 142.915 | 229154.000 | 0.053 | 0.667 |
| Std. Deviation | 1.408 | 2.965 | 20.410 | 109129.941 | 6.638 | 6.165 | 15905.713 | 0.006 | 0.059 |
| Valid N | 723 | 723 | 723 | 721 | 723 | 723 | 723 | 723 | 723 |

Note: *** $p < 0.01$, ** $p < 0.05$, * $p < 0.10$, two tailed test.

### 4.2. Regression Results, Mediation, and Moderation

#### 4.2.1. FOI and FOS's Impacts on Sustainability

Facing huge opportunities in the market, firms usually have two options: FOI and FOS. To test the influence of FOI and FOS on sustainability, this paper adopted linear regression models with the least squares method. Results are shown in Table 2.

**Table 2.** Regression results of FOI and FOS on sustainability.

| Dependent Variable: Sustainability | Model 1 | Model 2 | Model 3 | Model 4 |
|---|---|---|---|---|
| Independent variables: | | | | |
| FOI | | −0.088 ** | | −0.052 |
| FOS | | | 0.149 *** | 0.136 *** |
| Controls: | | | | |
| Return on assets | 0.212 *** | 0.200 *** | 0.209 *** | 0.202 *** |
| Debt to net worth ratio | −0.100 *** | −0.104 *** | −0.097 *** | −0.100 *** |
| Total employees | 0.187 *** | 0.182 *** | 0.191 *** | 0.188 *** |
| Executives proportion | −0.173 *** | −0.160 *** | −0.158 *** | −0.152 *** |
| Ratio of independent directors | −0.074 ** | −0.072 ** | −0.071 ** | −0.071 ** |
| Model-fitting metrics: | | | | |
| $R^2$ | 0.140 | 0.148 | 0.162 | 0.165 |
| Adjusted $R^2$ | 0.134 | 0.140 | 0.155 | 0.157 |
| F value | 23.398 *** | 20.663 *** | 23.123 *** | 20.149 *** |
| Observations | 723 | 723 | 723 | 723 |

Note: *** $p < 0.01$, ** $p < 0.05$.

Model 1 was the regression of sustainability with controls, which indicated that return on assets and total employees had positive effects on the sustainability, while the debt to net worth ratio, the proportion of executives, and the ratio of independent directors' impacts were negative. Model 2 examined the influence of FOI, whose regression coefficient was negatively significant. Therefore, Hypothesis 1 was verified. Model 3 checked the impact of FOS, the regression coefficient of which was positively significant, and thus Hypothesis 3 was confirmed. Model 4 considered FOI and FOS at the same time. Regression result manifested that the positive effect of FOS was still significant, while the negative impact of FOI was no longer significant. These statistical results showed that in the consumer goods industry under the background of demand-side reform, enterprises could not bring sustainable growth by focusing on innovation and increasing innovation investment, but rather focusing on sales and increasing sales force investment was conducive to the sustainable development of enterprises. From this conclusion, we believe that in the demand-side reform, enterprises in the consumer goods industry should tilt their resources to sales and quickly seize sales opportunities in the market, which is more conducive to the sustainable development of enterprises.

### 4.2.2. The Mediating Effect of FOS on the Effects of FOI on Sustainability

This study used the methods of Judd and Kenny [72] and Baron and Kenny [79] to examine the mediating effect of FOS. As shown in Model 2 and Model 4, FOI had negative effects on sustainability, and when FOS was controlled, its impact became insignificant. Consequently, there would be full mediating effects of FOS if FOI's impact on FOS were to be significant. The corresponding regression results are shown in Table 3.

**Table 3.** Regression results of FOI on FOS.

| Dependent Variable: FOS | Model 5 | Model 6 |
|---|---|---|
| Independent variable: Innovation | | −0.259 *** |
| Controls: | | |
| Return on assets | 0.021 | −0.014 |
| Debt to net worth ratio | −0.021 | −0.034 |
| Total employees | −0.032 | −0.044 |
| Executives proportion | −0.099 *** | −0.061 |
| Ratio of independent directors | −0.016 | −0.013 |
| Model-fitting metrics: | | |
| $R^2$ | 0.011 | 0.075 |
| Adjusted $R^2$ | 0.004 | 0.067 |
| F value | 1.539 | 9.615 *** |
| No. of Observations | 723 | 723 |

Note: *** $p < 0.01$.

Model 5 was the regression of FOS with controls, which revealed that the proportion of executives had an adverse effect on FOS. This means that the lower the proportion of executives, the more companies were inclined to focus their resources on sales when it came to demand-side reform. Model 6 was the regression of FOS on FOI, the result of which indicated that FOI negatively influenced FOS. Therefore, Hypothesis 2 was confirmed. Based on Hypothesis 1, Hypothesis 2, and Hypothesis 3, we can conclude that FOS acts as a full intermediary between FOI and sustainability. In other words, in demand-side reform, the reason why FOI in the consumer goods industry will show a negative correlation to sustainability is mainly because it inhibits FOS. FOI itself does not have a negative impact on sustainability, but its effect is slow, and it does not respond quickly to market changes. In contrast, FOS has a significant role in promoting corporate sustainability. In

market competition, if the enterprise cannot quickly seize the market opportunity, it will inevitably be surpassed by other enterprises, which will bring about a relative decline in the sustainable score.

The relationship between FOI, FOS, and sustainability is shown in Figure 2.

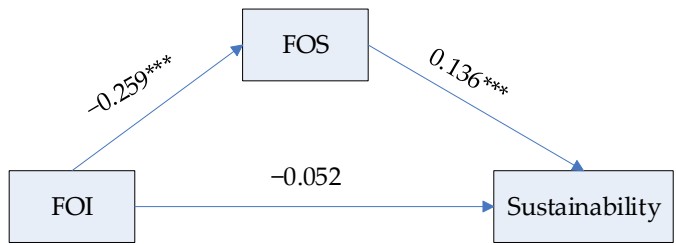

**Figure 2.** FOS's mediating effect between FOI and sustainability. Note: *** $p < 0.01$.

### 4.2.3. Firm Value's Moderating Effect Analysis

Enterprise value is an important variable used to measure the market value of a business, and it is closely associated with sustainability [80–82]. In this section, we examined its moderating influences on the effects of FOI and FOS on sustainability. The corresponding regression results are shown in Table 4.

**Table 4.** Moderating effect regression results.

| Dependent Variable: Sustainability | | Unstandardized Coefficients | | Standardized Coefficients | t | Sig. | Collinearity Statistics | |
|---|---|---|---|---|---|---|---|---|
| | | B | Std. Error | | | | Tolerance | VIF |
| Model 7 ($R^2$ = 0.165, F = 17.551, Sig < 0.001, N = 721) | Constant | 6.258 | 0.051 | | 123.539 | 0.000 | | |
| | FOI | −0.076 | 0.052 | −0.053 | −1.459 | 0.145 | 0.890 | 1.124 |
| | FOS | 0.190 | 0.050 | 0.135 | 3.792 | 0.000 | 0.925 | 1.082 |
| | Firm value | −0.012 | 0.055 | −0.008 | −0.214 | 0.830 | 0.770 | 1.298 |
| | Return on assets | 0.337 | 0.059 | 0.204 | 5.756 | 0.000 | 0.931 | 1.074 |
| | Debt to net worth ratio | −0.160 | 0.055 | −0.100 | −2.908 | 0.004 | 0.992 | 1.008 |
| | Total employees | 0.253 | 0.051 | 0.192 | 4.943 | 0.000 | 0.780 | 1.282 |
| | Executives proportion | −0.491 | 0.115 | −0.152 | −4.249 | 0.000 | 0.921 | 1.086 |
| | Ratio of independent directors | −0.110 | 0.053 | −0.071 | −2.050 | 0.041 | 0.983 | 1.018 |
| Model 8 ($R^2$ = 0.170, F = 14.591, Sig < 0.001, N = 721) | Constant | 6.262 | 0.052 | | 120.854 | 0.000 | | |
| | FOI | −0.082 | 0.054 | −0.057 | −1.516 | 0.130 | 0.824 | 1.214 |
| | FOS | 0.206 | 0.051 | 0.146 | 4.061 | 0.000 | 0.904 | 1.106 |
| | Firm value | 0.094 | 0.133 | 0.067 | 0.710 | 0.478 | 0.131 | 7.624 |
| | Firm value * FOI | −0.047 | 0.142 | −0.030 | −0.331 | 0.741 | 0.138 | 7.238 |
| | Firm value * FOS | 0.231 | 0.108 | 0.131 | 2.140 | 0.033 | 0.311 | 3.217 |
| | Return on assets | 0.319 | 0.059 | 0.193 | 5.397 | 0.000 | 0.910 | 1.099 |
| | Debt to net worth ratio | −0.159 | 0.055 | −0.100 | −2.904 | 0.004 | 0.992 | 1.008 |
| | Total employees | 0.264 | 0.066 | 0.201 | 3.986 | 0.000 | 0.461 | 2.170 |
| | Executives proportion | −0.476 | 0.116 | −0.147 | −4.115 | 0.000 | 0.915 | 1.093 |
| | Ratio of independent directors | −0.101 | 0.053 | −0.065 | −1.893 | 0.059 | 0.978 | 1.023 |

Model 7 was the regression of sustainability with firm value and other variables discussed above. Result indicated that firm value itself did not influence sustainability significantly. However, when the interaction items of firm value with FOI and FOS were added, the R-squares of the regression model changed significantly, and the interaction between firm value and FOS had a significant positive impact on sustainability, while the interaction between firm value and FOI had no significant impact on sustainability. This suggests that as FOS increases, the sustainability of large firms increases significantly more highly than that of SMEs. At the same time, large firms were not significantly different from SMEs in terms of the inhibitory effect of FOI on sustainability. Thus, Hypothesis 4 was not supported, while Hypothesis 5 was verified. In the context of demand-side reform, corporate value does not have a direct impact on sustainability and does not moderate the relationship between FOI and sustainability. Both large and small businesses are also trying to improve their sustainability. Small businesses are likely to grow and thrive after adopting the right strategies and tactics, and large businesses may decline if they are not well managed. However, large companies have a natural advantage in sales, marketing, and stakeholder relationships due to their abundant resources, capabilities, and corporate reputations, and if they invest more resources in FOS, they can do more with less.

The specific moderating effects are shown in Figure 3.

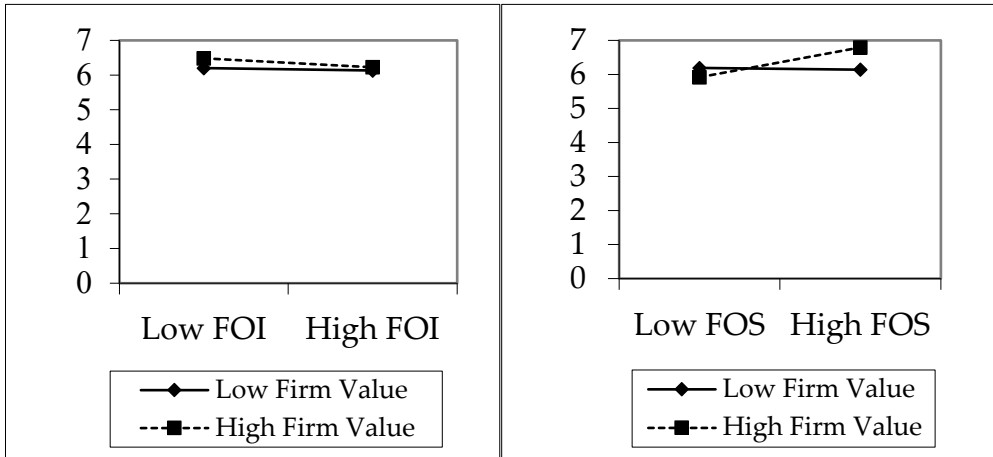

**Figure 3.** Firm value's moderating effects.

*4.3. Robustness Check*

4.3.1. Estimating with Robust Regression Method

Robust regression is a commonly used method of statistically robust estimation [83,84], the main idea of which is to modify the objective function in classical least squares regression, which is sensitive to outliers, to fit the structure of most of the data, while identifying potentially possible outliers, strong influence points, or structures that deviate from model assumptions. When the error follows a normal distribution, its estimate is almost as good as the least squares estimate, and when the least squares estimation condition is not met, the result is better than the least squares estimate. We used the robust estimator of variance in this study to conduct the robustness check first, which was robust to some types of misspecifications so long as the observations were independent. The results are shown in Table 5.

**Table 5.** Robustness check results with robust regression method.

| Dependent Variable | Sustainability | Sustainability | FOS | Sustainability | Sustainability |
|---|---|---|---|---|---|
| Independent variable: | | | | | |
| FOI | −0.088 ** | | −0.259 *** | −0.052 | −0.057 |
| FOS | | 0.149 *** | | 0.136 *** | 0.146 *** |
| Moderator: | | | | | |
| Firm value | | | | | 0.067 |
| Interactions: | | | | | |
| Firm Value * FOI | | | | | −0.030 |
| Firm Value * FOS | | | | | 0.131 *** |
| Controls: | | | | | |
| Return on assets | 0.200 *** | 0.209 *** | −0.014 | 0.202 *** | 0.193 *** |
| Debt to net worth ratio | −0.104 *** | −0.097 *** | −0.034 *** | −0.100 *** | −0.100 *** |
| Total employees | 0.182 *** | 0.191 *** | −0.044 * | 0.188 *** | 0.201 *** |
| Executives proportion | −0.160 *** | −0.158 ** | −0.061 ** | −0.152 *** | −0.147 *** |
| Ratio of independent directors | −0.072 ** | −0.071 ** | −0.013 | −0.071 * | −0.065 * |
| $R^2$ | 0.148 | 0.162 | 0.075 | 0.165 | 0.171 |
| F Value | 21.392 *** | 24.038 *** | 8.854 *** | 21.063 *** | 19.603 *** |
| No. of Observations | 723 | 723 | 723 | 723 | 721 |

Note: *** $p < 0.01$, ** $p < 0.05$, * $p < 0.10$.

As can be seen from Table 5, FOI had a significant negative impact on sustainability and FOS had a significant positive impact on sustainability. After considering both FOI and FOS, the negative impact of FOI on sustainability became insignificant. Moreover, FOI also had a significant negative effect on FOS. In terms of the moderating effect, the interaction between enterprise value and FOI did not have a significant impact on sustainability, but the interaction between enterprise value and FOS had a significant and positive impact on sustainability. These results are consistent with the above estimates using the least squares method, which proves that this study has good robustness.

4.3.2. Replacing ESG Scores with Environmental Performance

In the study above, we used ESG scores to measure sustainability. As a measure of sustainability, the most important component of ESG is the environmental score. In early sustainability studies, environmental issues were dominant. Therefore, the environmental score was used instead of the ESG score as another way to conduct the robustness test in this research. The results are shown in the Table 6.

As can be seen from Table 6, FOI still had a significant negative impact on sustainability, and FOS had a significant positive impact on sustainability. However, after considering both FOI and FOS, the negative impact of FOI on sustainability and the positive impact of FOS on sustainability remained significant. Furthermore, the negative impact of FOI on FOS was still significant. In terms of the moderating effect, the interaction between enterprise value and FOI did not have a significant impact on sustainability, but the interaction between enterprise value and FOS had a significant positive impact on sustainability. In addition to this, firm value itself had a significant positive impact on sustainability. That is, if environmental performance is used as a measure of sustainability, large enterprises may perform better and have stronger sustainability than small enterprises. Overall, these conclusions are still consistent with the above estimates using ESG as a sustainability measurement variable, thus demonstrating the robustness of this study.

**Table 6.** Robustness check results with variable substitution method.

| Dependent Variable | Sustainability | Sustainability | FOS | Sustainability | Sustainability |
|---|---|---|---|---|---|
| Independent variable: | | | | | |
| FOI | −0.182 *** | | −0.259 *** | −0.130 *** | −0.131 *** |
| FOS | | 0.236 *** | | 0.204 *** | 0.228 *** |
| Moderator: | | | | | |
| Firm value | | | | | 0.234 ** |
| Interactions: | | | | | |
| Firm Value * FOI | | | | | −0.012 |
| Firm Value * FOS | | | | | 0.210 *** |
| Controls: | | | | | |
| Return on assets | 0.007 | 0.027 | −0.014 | 0.010 | −0.021 |
| Debt to net worth ratio | −0.009 | −0.005 | −0.034 | −0.002 | −0.001 |
| Total employees | 0.202 *** | 0.218 *** | −0.044 | 0.211 *** | 0.181 *** |
| Executives proportion | −0.016 | −0.020 | −0.061 | −0.004 | 0.005 |
| Ratio of independent directors | −0.020 | −0.019 | −0.013 | −0.017 | −0.015 |
| $R^2$ | 0.083 | 0.107 | 0.075 | 0.122 | 0.145 |
| F Value | 10.860 *** | 14.283 *** | 9.615 *** | 14.171 *** | 12.007 *** |
| No. of Observations | 723 | 723 | 723 | 723 | 721 |

Note: *** $p < 0.01$, ** $p < 0.05$.

## 5. Discussions

Most of the existing research focuses on environmental sustainability issues while ignoring sustainability at the social and corporate governance levels. This has created confusion among consumers and businesses about how to manage sustainability [85–87]. Especially for the consumer goods industry, which faces relatively fewer environmental regulations than industries with more prominent environmental problems such as heavily polluted industries, sustainability at the social and corporate governance levels is more important. The relationship theory of environmental regulation and innovation and performance in the Porter hypothesis [88] has good convincing power in high-pollution and energy-consuming industries, but it does not fully apply to consumer industries under the COVID-19 and the demand-side reform, as there are fewer environmental regulations. The research in this paper confirms the role of FOS in promoting sustainability, which is consistent with Melander and Pazirandeh [43] and Ferrer et al. [38] conclusions about the impact of sales and marketing relationships on sustainability. However, the research in this paper also shows that the impact of FOI on sustainability is not significant under the influence of limited resources and short-term opportunities in the market, and even negative correlation will occur, mainly through FOS. This is a useful complement to the findings of studies such as Forcadell et al. [19] and Desiana et al. [89], namely, that innovation in general contexts significantly contributes to sustainability.

### 5.1. The Relationship between FOI, FOS, and Sustainability

This paper examines the innovation and market strategic decision-making of consumer goods enterprises under the dual factors of existential crisis and government subsidies and their sustainability impact, which is significantly different from the traditional general scenario of corporate innovation and sustainability research. The outbreak of the new crown pneumonia epidemic at the end of 2019 has hit the economy of China and the world hard. With the economic downturn and epidemic control, the income and consumption capacities of residents have also been greatly affected and have directly endangered the

survival and development of enterprises in the consumer goods industry. In order to help enterprises tide over the difficulties, the Chinese government has implemented the demand-side reform policy nationwide through cooperation with local governments, large shopping malls, well-known enterprises, etc., to jointly provide consumer subsidies to stimulate the consumer demand and market recovery.

In this context, the relationship between innovation decision-making, the sales behavior, and the sustainability of enterprises has taken on new characteristics. First, we found that FOI was negatively correlated with corporate sustainability. The results of Pearson's correlation coefficient, general linear regression, robust regression, and other methods all confirm this conclusion. This is contrary to most previous studies. In the common scenario, through FOI, companies can improve technology and increase productivity and service levels, which are conducive to the sustainable development of enterprises [26–28].

Nevertheless, the COVID-19 pandemic and the advent of demand-side reform have changed the relationship. Under the influence of the new crown pneumonia epidemic, enterprises are facing greater pressure to survive. Due to the continuous outbreak of epidemic closure, travel control and the decline in residents' consumption capacity, the survival of enterprises in the consumer goods industry is particularly serious, and some shopping malls and small- and medium-sized enterprises have even closed their doors or gone bankrupt. Previous studies have shown that the impact of innovation on business performance is often a long-term and slow process that requires many resources [12,62]. Under the influence of the new crown pneumonia epidemic, the survival pressure of enterprises is relatively large, and if they continue to invest too many resources in research and development, the efficiency of resource use would be lower than other companies who focus their resources on sales and market. Therefore, the sustainability of the business may also be lower.

On the other hand, as previously found in the literature [43], the results of our data analysis also show that FOS has a significant positive role in promoting corporate sustainability. Consequently, companies that focus on sales can lead to better sustainability than companies that focus on innovation. Especially for the consumer goods industry, where the company's products are directly oriented to the market and customers, FOS is very important for the survival and sustainable development of the enterprise. Under the background of demand-side reform, there are greater opportunities in the market, and enterprises can better promote survival and sustainable development through FOS than FOI.

As to the relationship between FOI and FOS, many studies indicate that innovation can boost sales and markets in the absence of mutually exclusive resource competition or low-level resource constraints [47–49]. However, under the impact of the COVID-19, the production and operation of enterprises are facing difficulties, and resources have become more limited. As a result, companies need to make trade-offs between FOI and FOS, because investing resources in innovation means that it is no longer possible to invest in sales, and vice versa. The statistical analysis results of the data in this paper also prove this negative relationship between FOI and FOS.

Finally, the mediation effect test process and results once again confirm the above analysis. From the perspective of correlation, there is a negative correlation between FOI and sustainability. However, in practice, this negative correlation is not significant after excluding the effects of FOS. The reason is that the negative impact of FOI on sustainability is mainly due to its resource competition with FOS. In the consumer industry, under the new crown pneumonia epidemic and demand-side reform, enterprises need to use FOS to ensure their basic viability. Especially for small- and medium-sized enterprises, only by getting through the survival difficulties first can they better seek innovation and development. These findings not only theoretically enrich the relationship between FOI, FOS, and sustainability, but also have positive implications for the COVID-19 response in other industries.

### 5.2. The Impacts of Firm Value

Maximizing firm value is the main goal of financial management and an important basis for the board of directors to evaluate managers [90–92]. In this study, we used the method of the Wind database company who calculated firm value from the sum of the value of equity and the value of liabilities. According to this method, companies with higher enterprise value are generally stronger in size and profitability and tend to have stronger viability and sustainability. The descriptive statistical results in this paper also confirm the judgement that corporate value and sustainability have a significant positive correlation. In fact, the statistics of our study also manifest that corporate value by itself does not have a significant impact on sustainability. The reason for the positive relationship between it and sustainability is, on the one hand, that it is closely related to factors such as the asset/liability ratio, the total number of employees, the proportion of executives, the proportion of independent directors, etc., all of which have a significant impact on corporate sustainability. On the other hand, in the context of demand-side reform, corporate value may moderate the positive effect of FOS on sustainability.

For firms of the consumer goods industry during COVID-19, demand-side reform is a creative temporary subsidy mechanism undertaken by the Chinese government. Prior to the implementation of demand-side reform, the consumer goods industry had already been severely affected by the COVID-19 pandemic. Seizing the demand-side reform opportunities provided by the government and making in-depth efforts in marketing and promotion is an important way for enterprises to remove inventory and increase sales revenue from their main business. In this regard, there are significant differences in the impact of FOS strategies on sustainability between companies with large market value and those with small market value. The statistical analysis results of this paper using consumer goods industry data show that the market value of firms would positively moderate the relationship between FOS and sustainability. That is, the higher the market value of a business, the stronger the impact of FOS on sustainability.

The reason is that the value of the enterprise is not only the embodiment of the value of the enterprise's own assets and liabilities, but it also reflects the true views of the public and the market on the capital strength and brand influence of the enterprise. Compared with enterprises with low market value, enterprises with high market value have more advantages in terms of customer recognition, social public relations, and market influence, and the FOS strategy can play a better role. Many news reports have also shown that in the new crown pneumonia epidemic, large- and medium-sized enterprises with high market value are the main participants in demand-side reform, and together with government departments, they have provided many special subsidies to consumers and achieved a rebound in product sales revenue [93–95]. In contrast, small- and medium-sized enterprises, due to limited resources, capabilities, and weak social networks, even if they have been tilted in terms of FOS strategy, still have disadvantages over large enterprises.

In the statistical analysis of this paper, we also find that the moderating effect of firm value on the relationship between FOI and sustainability is insignificant. Combined with the above conclusion that the FOI strategy does not have a significant impact on sustainability after controlling FOS and other variables, it shows that neither companies with high market value nor those with low market value may significantly influence sustainability through FOI. In the context of COVID-19 and demand-side reform, large enterprises with high market value and small- and medium-sized enterprises with low market value are facing a certain degree of resource shortage. Therefore, when many favorable opportunities suddenly arise in the market environment, by shifting resources and strategies from FOI to FOS, both kinds of firms may reduce the negative correlation between FOI and sustainability and promote their sustainability. However, it should be noted that this strategic shift is not conducive to corporate innovation and high-quality development of the consumer goods industry in the long run. In China, the transformation and upgrading of the consumer goods industry is the trend of the times. After COVID-19, resource constraints will be eased as the economy improves, and companies should insist

on using innovation as the main way to grow profits or at least to maintain a modest investment in innovation to provide better products and services.

## 6. Conclusions

In the context of demand-side reform, opportunities from the market side have increased, and sales growth has become more attractive to enterprises. On this background, should companies focus on sales or innovation and how would they impact sustainability? This paper analyzes the strategic choices of enterprises and the moderating effects of corporate value and draws three main conclusions. First, in the context of demand-side reform, for the consumer goods industry, the impact of FOI on corporate sustainability is negative, and the impact of FOS on corporate sustainability is positive. From the perspective of improving the sustainable development ability of enterprises, it is more appropriate for enterprises to choose FOS rather than FOI. Second, FOI does not have a direct negative impact on the sustainability of the business. The reason why it has a significant negative correlation with sustainability is mainly because it has a significant negative impact on FOS, which indirectly inhibits the sustainability of the enterprise. For the consumer goods industry under the demand-side reform, because the opportunities in the market are short and huge, and the changes in innovation are slow and too late to respond, the sustainability of the enterprise can be promoted by focusing on sales. Third, although corporate value does not significantly moderate the relationship between FOI and sustainability, it has a significant positive moderating effect on the positive relationship between FOS and sustainability. In the context of demand-side reform, large enterprises can obtain greater sustainability improvements than small enterprises through FOS.

There are two main limitations to this study. First, considering the availability of data, the sample companies selected in this study are all listed companies. However, for the consumer goods industry, there are more small unlisted businesses. This paper fails to study how these companies should respond to the market opportunities brought about by demand-side reform, and it is not known whether the research conclusions apply to these enterprises. Second, under the influence of the new crown pneumonia epidemic in 2019, all walks of life are generally facing the problem of declining demand and insufficient consumption. When the government makes new policies and boosts consumption, should companies continue to keep investing in innovation or focus on market sales? Obviously, the consumer goods industry is most directly affected by the government's demand-side reform policies, which is the main reason why the consumer goods industry was selected as a sample in this study. However, for other industries, are there similar conclusions to this paper in terms of FOI and FOS corporate decisions? This remains to be explored. We hope that subsequent studies will be able to explore in depth in both above areas. More in-depth and more general, if possible, we hope that follow-up research will be able to explore the relationship among market opportunity intensity, FOI, and FOS decision-making, and corporate sustainability.

**Author Contributions:** Conceptualization, G.X. and S.W.; methodology, G.X.; software, G.X.; validation, G.X. and S.W.; writing—original draft preparation, G.X.; writing—review and editing, S.W. and Z.S.; supervision, S.W.; project administration, S.W. and Z.S.; funding acquisition, S.W. and Z.S. All authors have read and agreed to the published version of the manuscript.

**Funding:** This research was funded by Philosophy and Social Science Project of Guizhou Province, China (Grant number: 21GZQN13) and the Key Project of Humanities and Social Sciences Research, Department of Education, Guizhou Province, China (Grant number: 2022ZD008).

**Data Availability Statement:** All the data used in this research can be downloaded at https://www.wind.com.cn/en/default.html since 1 January 2022.

**Acknowledgments:** We appreciate the comments of the anonymous reviewers.

**Conflicts of Interest:** The authors declare no conflict of interest.

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
