# Peer review of "Focus on Innovation or Focus on Sales? The Influences of the Government of China’s Demand-Side Reform during COVID-19 and Their Sustainability Consequences in the Consumer Products Industry"

_sustainability, doi:10.3390/su142013348_

Round 1

Reviewer 1 Report

The issues raised are topical and interesting. The article presents all important issues for the scientific text: theoretical, methodical and empirical. The scope of literary studies is satisfactory. Research results can be an important source for comparative research.

I am requesting the following changes:

1. the division of the current content of the Conclusions into three parts.

2. the first of these parts should be the Conclusions, in which the results will be summarized.

3. the second part should be the Discussion, which only to a narrow extent included in the text, the Authors should refer to the results of other studies a little more, comparing them with their own.

4. the third part should concern only the issue of limitation of the application with justification.

Author Response

Dear Reviewer,

Thank you very much for your attentive and helpful comments. Another reviewer also made a similar comment on the conclusion section. Based on your comments, we have adjusted the conclusion section accordingly, and the adjusted conclusion part is as follows:

  1. Conclusions

In the context of Demand-Side Reform, opportunities from the market side have increased, and sales growth has become more attractive to enterprises. On this back-ground, should companies focus on sales or innovation and how would they impact sustainability? This paper analyzes the strategic choices of enterprises and the moderating effect of corporate value, and obtains three main conclusions. First, in the context of Demand-Side Reform, for the consumer goods industry, the impact of FOI on corporate sustainability is negative, and the impact of FOS on corporate sustainability is positive. From the perspective of improving the sustainable development ability of enterprises, it is more appropriate for enterprises to choose FOS than FOI. Second, FOI does not have a direct negative impact on the sustainability of the business. The reason why it has a significant negative correlation with sustainability is mainly because it has a significant negative impact on FOS, which indirectly inhibits the sustainability of the enterprise. For the consumer goods industry under the Demand-Side Reform, be-cause the opportunities in the market are short and huge, and the changes in innovation are slow and too late to respond, the sustainability of the enterprise can be pro-moted by focusing on sales. Third, although corporate value does not significantly moderate the relationship between FOI and sustainability, it has a significant positive moderating effect on the positive relationship between FOS and sustainability. In the context of Demand-Side Reform, large enterprises can obtain greater sustainability improvements than small enterprises through FOS.

Most of the existing research focuses on environmental sustainability issues, while ignoring sustainability at the social and corporate governance levels. This has created confusion among consumers and businesses about how to manage sustainability [93-95]. Especially for the consumer goods industry, which faces relatively fewer environmental regulations than industries with more prominent environmental problems such as heavily polluted industries, sustainability at the social and corporate governance levels is more important. The relationship theory of environmental regulation and innovation and performance in the Porter hypothesis [96] has good convincing power in high-pollution and energy-consuming industries, but it does not fully ap-ply to consumer industries under the COVID-19 and the Demand-Side Reform as there are fewer environmental regulations. The research in this paper confirms the role of FOS in promoting sustainability, which is consistent with Melander and Pazirandeh [43] and Ferrer et al. [38] 's conclusions about the impact of sales and marketing relationships on sustainability. However, the research in this paper also shows that the impact of FOI on sustainability is not significant under the influence of limited re-sources and short-term opportunities in the market, and even negative correlation will occur, mainly through FOS. This is a useful complement to the findings of studies such as Forcadell et al. [19] and Desiana et al. [97] that innovation in general contexts significantly contributes to sustainability.

There are two main limitations to this study. First, considering the availability of data, the sample companies selected in this study are all listed companies. However, for the consumer goods industry, there are more small unlisted businesses. This paper fails to study how these companies should respond to the market opportunities brought about by Demand-Side Reform, and it is not known whether the research conclusions apply to these enterprises. Second, under the influence of the new crown pneumonia epidemic in 2019, all walks of life are generally facing the problem of declining demand and insufficient consumption. When the government makes new policies and boosts consumption, should companies continue to keep investing in innovation or focus on market sales? Obviously, the consumer goods industry is most directly affected by the government's Demand-Side Reform policies, which is the main reason why the consumer goods industry was selected as a sample in this study. But for other industries, are there similar conclusions to this paper in terms of FOI and FOS corporate decisions? This remains to be explored. We hope that subsequent studies will be able to explore in depth in both above areas. More in-depth and more general, if possible, we hope that follow-up research will be able to explore the relationship among market opportunity intensity, FOI and FOS decision-making, and corporate sustainability.

Thanks Again!

 Best Regards!

                                                      The Authors

Reviewer 2 Report

Paper 1929797 review to Sustainability – Focus on innovation or focus on sales? The influences of China government’s Demand Side Reform in COVID-19 and their sustainability consequences in consumer products industry

All issues raised in this review can be considered to be minor reviews.

General considerations

The subject of business sustainability is very relevant and current, and this article is an asset to highlight companies that are committed to survive in the business markets in which they operate. The article is very well structured, the contents explained with a good level of depth and well-articulated with each other. The data collection method and the analysis of the results obtained are presented in a perceptible and complete way. All issues raised in this review can be considered to be minor reviews. The limitations of the study performed are well explained and well-founded.

1.   Structure

The structure of the article is well elaborated, but some flaws detected in the numbering of the sections and subchapters presented, namely in lines:

·         430 - The numbering of this subsection should be 4.3.1;

·         450 - The numbering of this subsection should be 4.3.2.

2.   Title, Abstract and Keywords

·         The title is appealing to readers.

·         The abstract is well constructed. The main research question, the objectives and the development of the theme are clearly pointed out. But, whenever a term with its acronym is used, the first letters of each word must be capitalized.

·         The keywords are adequate.

3.      Figures and tables

The figures and tables are all well numbered, and have good visual quality. But the areas of the graphs in Figure 2 are too large, taking up too much space in the article. Authors should reduce them to much smaller proportions, but without compromising the good visual quality they present.

4.      Grammar, spelling and syntax issues

The whole article it's well written in terms of grammar and spelling. But there were identified some aspects that should be improved/corrected, namely:

·         Whenever a term with its acronym is used, the first letters of each word must be capitalized;

·         In line 133 - missing 1 full stop after the reference;

·         In lines 211 and 233 - there are 2 endpoints after the reference;

·         In line 251 - the acronym ESG must be accompanied by its full meaning, as this is the first time it is mentioned in the article;

·         In line 253 - the acronym FTSE must be accompanied by its full meaning, as this is the first time it is mentioned in the article;

·         In line 236 - there is 1 period and 1 comma followed by each other, after the reference.

5.      Semantic and technical issues

The entire article is very well explained. The issues are explained very clearly and the concepts and ideas are very well articulated between themselves. The data collection method is explained clearly and objectively. However before Table 1, the sustainability variable lacks an explanation of the numerical concept it represents, that is, a more detailed explanation of how it is calculated.

6.      References

The list of references is well prepared, the number of references is appropriate to the depth of the theme's approach in the article. The references are strong in the scope of this investigation. But throughout the text, the authors indicate the references in two ways simultaneously: with year-names and with numbers; and if the authors must reference in the first way, they must use the term "et al." in italics and without the period before the last parentheses, as they do so many times.

Author Response

Dear Reviewer,

Thank you very much for your attentive and helpful comments. We are very sorry for the editing errors in the article. Based on your comments, we have made changes accordingly:

1.Structure

The structure of the article is well elaborated, but some flaws detected in the numbering of the sections and subchapters presented, namely in lines:

  • 430 - The numbering of this subsection should be 4.3.1; (Revised, now in line 484)
  • 450 - The numbering of this subsection should be 4.3.2. (Revised, now in line 505)

2.Title, Abstract and Keywords

  • The title is appealing to readers.
  • The abstract is well constructed. The main research question, the objectives and the development of the theme are clearly pointed out.But, whenever a term with its acronym is used, the first letters of each word must be capitalized.

This section deals with two main phrases: Focus On Sales and Focus On Innovation. Adjustments have been made accordingly. In addition, considering that Demand-Side Reform is a proper noun, we have capitalized its initials.

  1. Figures and tables

The figures and tables are all well numbered, and have good visual quality. But the areas of the graphs in Figure 2 are too large, taking up too much space in the article. Authors should reduce them to much smaller proportions, but without compromising the good visual quality they present.

We have reduced the size of this graph. Since a theoretical model diagram was added at the end of the Literature Review and Research Hypothesis section, this diagram is currently Figure 3, in line 525.

  1. Grammar, spelling and syntax issues

The whole article it's well written in terms of grammar and spelling. But there were identified some aspects that should be improved/corrected, namely:

  • Whenever a term with its acronym is used, the first letters of each word must be capitalized; (We checked the whole paper and revised this issue, which can be viewed in revision mode of the revised version)
  • In line 133 - missing 1 full stop after the reference; (Revised, now in line 146)
  • In lines 211 and 233 - there are 2 endpoints after the reference; (Revised, now in lines 228 and 249)
  • In line 251 - the acronym ESG must be accompanied by its full meaning, as this is the first time it is mentioned in the article; (Added, now in line 301)
  • In line 253 - the acronym FTSE must be accompanied by its full meaning, as this is the first time it is mentioned in the article; (Added, now in line 303)
  • In line 236 - there is 1 period and 1 comma followed by each other, after the reference. (The period is deleted, now in line 251)

5.Semantic and technical issues

The entire article is very well explained. The issues are explained very clearly and the concepts and ideas are very well articulated between themselves. The data collection method is explained clearly and objectively. However before Table 1, the sustainability variable lacks an explanation of the numerical concept it represents, that is, a more detailed explanation of how it is calculated.

We have added a description of the value process for the sustainability variable as follows: To be specific, C has the lowest grade and is assigned the value of 1. AAA has the highest rating and is assigned the value of 9. Among the other grades, CC grade is as-signed the value of 2, CCC grade is assigned the value of 3, B grade is assigned the value of 4, BB grade is assigned the value of 5, BBB grade is assigned a value of 6, A grade is assigned the value of 7, and AA grade is assigned the value of 8.

6.References

The list of references is well prepared, the number of references is appropriate to the depth of the theme's approach in the article. The references are strong in the scope of this investigation. But throughout the text, the authors indicate the references in two ways simultaneously: with year-names and with numbers; and if the authors must reference in the first way, they must use the term "et al." in italics and without the period before the last parentheses, as they do so many times.

We have revised the reference citation format for the full text and can be viewed in the revised version.

Finally, we would like to thank the reviewer for the careful reading and time again, again, and again!

Best Regards!

                                                      The Authors

Reviewer 3 Report

I enjoyed reading the paper as the theme was relevant. Please find below my concerns so far.

Introduction.

Problematic is expressed in a simple way (that is good). However, what is the real RQ: As it is presented in the article " To seize this market opportunity, should firms focus on innovation or sales? How would they affect sustainability, and which one is more important? This is the first question of the research". As to me there are 3 questions here!. Please clarify this point.

I would also add more references as it lacks the second part of the introduction. Explaining more deeply the contributions (and not just synthetizing the results) should be done.

Literature review

I would add some sub-levels in the literature review, allowing the reading to see more clearly the concepts defined and the hypotheses linked. We need definitions of concepts (those are missing) as well as a clear positioning in regards to previous literature. As it is, the section is a little bit too short. A conceptuel model should be presented at the end of the literature review (with hypotheses).

Data and methodology

Some explanation of the choice of data set (why this one, why it helps the research, and why secondary data are used - with references) has to be added.

Sub-section 3.3. needs references!

All along the presentation of results sentences like this one (Firm value is a representative variable used to measure the size of a business.) are presented. However, any assumption has to be conforted by at least 1 reference. Please read carefully that section and adjust.

Once again in 4.2.1 you write "Robust regression is a method of statistically robust estimation". Important statement without reference.

Discussion of the results is missing. How one should understand your results (what does it mean!). Discussion is not just presenting the results and positioning them in regards to previous literature. Once said that, what are the contributions of the research?

I would suggest add a discussion section and reorganise the conclusion with a contribution section and just a small paragraph for conclusion

Author Response

Dear Reviewer,

Thank you very much for your detailed and enlightening comments. Your comments are the most substantive of the three reviewers, and we have the most revisions this time. In fact, taking the advice of two other reviewers, it took us only two hours to revise the article. But after receiving your valuable comments, we revised the article for nearly ten days. When we finished revising it, we reviewed this article and felt that the overall quality had been greatly improved.

In the introduction, we clarified the research question, explained the contributions more deeply and added some new references. The revised parts are shown below:

Facing global and civil economic recession brought by COVID-19, China’s house-hold consumption suffered a lot in 2020. Many small and medium firms of consumer products industry encountered poor sales and survival dilemma. The government implemented a policy called Demand-Side Reform in the end of 2020 to help firms to pull through, by adopting multitudinous measures covering political, economic, financial, social, etc. One of the most attractive measures for firms is the consumption coupon provided directly to consumers by the government, which cut the actual payment price down from consumers side through public finance expenditures. To seize this market opportunity, should companies focus on innovation or sales, and what impacts do they have on sustainability? This is the main question of this study. Besides, considering the significance of FOI and its probable impact on FOS, this paper would further study its direct effect to sustainability and indirect effect through FOS. In addition, firm value is introduced as a moderator to explore its influence on the relationship between FOI, FOS, and sustainability.

Motivated by the relationship between FOI, FOS, and their impacts on sustainability in consumer products industry under the background of China’s Demand-Side Reform, this paper applies public data to empirically address the above questions. To be specific, we first conduct a correlation analysis and hierarchical regression model to observe the influence of FOI and FOS to sustainability. Results manifest that FOI negatively affect sustainability, while FOS’s impact is positive. Then, the mediation effect test of FOS between FOI and sustainability is examined, which indicate that FOS fully mediates the adverse effect of FOI on sustainability. To go further, we explore and find that firm value positively moderates the relationship between FOS and sustainability through moderating effect analysis method. Nevertheless, firm value’s moderating effect between FOI and sustainability is insignificant. Two kinds of robustness tests are conducted to make the results more convincing.

This paper contributes to the current literature in several ways. First, we found that for the consumer goods industry, when short-term and huge promotional opportunities arise in the market environment, FOI has a negative impact on the sustainability of the enterprise, while FOS can promote the sustainable improvement of the enterprise. At this point, FOS is more beneficial to the sustainability of the business than FOI. This new context and conclusion challenges traditional research to emphasize that companies should focus on innovation rather than sales [9-10]. Second, further research shows that the negative impact of FOI on sustainability is entirely mediated by FOS. This result indicates that FOI itself does not compromise sustainability, and the main reason for its negative impact on sustainability in the context of Demand-Side Reform is to crowd out the resources needed by FOS, which has a significant ad-vantage in seizing market opportunities to improve sustainability. This finding has implications for differentiating industries and contexts to study the impact of innovation on sales and sustainability [5,8]. Third, this paper further explores the role of firm value in the relationship between FOI, FOS, and corporate sustainability, and finds that in Demand-Side Reform, large enterprises are more able to seize opportunities and improve corporate sustainability through FOS than small enterprises, but there is no significant difference between the two in terms of FOI and sustainability. This result enriches research on the impact of firm value on innovation, sales and sustainability and opens new explorations in the context of Demand-Side Reform [12].

The rest of this paper is organized as follows: Section 2 presents a brief literature review related to innovation, sales and sustainability and proposes corresponding hypotheses. Section 3 illustrates the data collection process and research methodology. Section 4 shows the statistical results and discusses the findings. Section 5 explores the insights, and concludes the paper with a discussion on research limitations and future research directions.

In the literature review, we added some sub-levels in the literature review, allowing the reading to see more clearly the concepts defined and the hypotheses linked. We also supplemented the definitions of concepts as well as a clear positioning in regards to previous literature. Besides, a conceptual model is presented at the end of the literature review with hypotheses.

In the data and methodology, we added the explanation of the choice of data set (why this one, why it helps the research, and why secondary data are used - with references). The following is the revised version:

The data sample of this study is mainly from the consumer products industry enterprises listed on China's A-share market, including food and main supplies retail, food, beverages and tobacco, household and personal products, automobiles and auto parts, durable consumer goods and clothing, consumer services, media, retail and other industries, which comprehensively covers the daily consumption and optional consumption in the Wind company’s industry classification system, which can better reflect the overall situation of China's consumer market. Wind is a comprehensive data-base company that collects data from Listed Companies in China, and its corporate financial data and ESG data are wide used in academic research [71-73]. Considering the authority and verifiability of the data sources, we used the Wind database's second-hand data in this study. As of August 31, 2021, the total number of A-share listed consumer products industry enterprises on the Shanghai Stock Exchange and the Shenzhen Stock Exchange was 920. Among them, 885 firms were included in the ESG (Environmental, Social and Governance) rating of Sino- Securities Index, 138 firms were included in the Wind ESG rating, and 123 firms were included in the FTSE (Financial Times Stock Exchange) Russell ESG rating list. Considering the comprehensiveness of the data selection, this study uses 885 consumer products industry enterprises included in the ESG rating list of Sino- Securities Index as the research object, and after eliminating the missing values, 723 sample companies are selected (two firms which only have missing value in firm value are kept), and the specific data comes from the financial report and the Sino- Securities ESG Index of listed companies in Wind database.

For Sub-section 3.3. and other areas that require reference support, for example, the mediation effect test, the moderating effect test, and the robust regression method, we have added citations accordingly. The following is the revised version of Sub-section 3.3:

Focusing on the relationship between FOI, FOS and sustainability in the consumer products industry under Demand-Side Reform, and exploring the moderating role of enterprise value, this paper proposes to use classic methods including general linear regression, mediation effect test [85], moderation effect test [86], and robustness test [77], and divide the empirical research into four parts: (1) test the impacts of FOI and FOS on the sustainability of enterprises; (2) To test the impacts of FOI on the sustaina-bility of the enterprise through FOS; (3) Test the moderating effect of enterprise value on FOI and FOS to sustainability; (4) Robustness tests are carried out by estimating with robust regression method and replacing environmental ratings to ESG scores.

In addition, we have added a special chapter to fully discuss the conclusions of the research in this paper and their significance, which is shown below:

  1. Discussions

5.1. The Relationship between FOI, FOS, and Sustainability

This paper examines the innovation and market strategic decision-making of consumer goods enterprises under the dual factors of existential crisis and government subsidies and their sustainability impact, which is significantly different from the traditional general scenario of corporate innovation and sustainability research. The out-break of the new crown pneumonia epidemic at the end of 2019 has hit the economy of China and the world hard. With the economic downturn and epidemic control, the in-come and consumption capacity of residents have also been greatly affected, and directly endangered the survival and development of enterprises in the consumer goods industry. In order to help enterprises tide over the difficulties, the Chinese government has implemented the Demand-Side Reform policy nationwide, through cooperation with local governments, large shopping malls, well-known enterprises, etc., to jointly provide consumer subsidies to stimulate the consumer demand and market recovery.

In this context, the relationship between innovation decision-making, sales behavior and sustainability of enterprises has taken on new characteristics. First, we found that FOI was negatively correlated with corporate sustainability. The results of Pearson's correlation coefficient, general linear regression, robust regression, and other methods all confirm this conclusion. This is contrary to most previous studies. In the common scenario, through FOI, companies can improve technology, increase productivity and service levels, which are conducive to the sustainable development of enterprises [26-28].

Nevertheless, the COVID-19 pandemic and the advent of Demand-Side Reform have changed the relationship. Under the influence of the new crown pneumonia epi-demic, enterprises are facing greater pressure to survive. Due to the continuous out-break of epidemic closure, travel control and the decline in residents' consumption capacity, the survival of enterprises in the consumer goods industry is particularly serious, and some shopping malls and small and medium-sized enterprises have even closed their doors or gone bankrupt. Previous studies have shown that the impact of innovation on business performance is often a long-term and slow process that re-quires many resources [62-63]. Under the influence of the new crown pneumonia epi-demic, the survival pressure of enterprises is relatively large, and if they continue to invest too much resources in research and development, the efficiency of resource use would be lower than other companies who focus their resources on sales and market. Therefore, the sustainability of the business may also be lower.

On the other hand, as previously found in the literature [43], the results of our da-ta analysis also show that FOS has a significant positive role in promoting corporate sustainability. Consequently, companies that focus on sales can lead to better sustain-ability than companies that focus on innovation. Especially for the consumer goods industry, where the company's products are directly oriented to the market and customers, FOS is very important for the survival and sustainable development of the enterprise. Under the background of Demand-Side Reform, there are greater opportunities in the market, and enterprises can better promote survival and sustainable development through FOS than FOI.

As to the relationship between FOI and FOS, many studies indicate that innovation can boost sales and markets in the absence of mutually exclusive resource competition or low-level resource constraints [47-49]. However, under the impact of the COVID-19, the production and operation of enterprises are facing difficulties, and re-sources have become more limited. As a result, companies need to make trade-offs between FOI and FOS, because investing resources in innovation means that it is no longer possible to invest in sales. Vice versa. The statistical analysis results of the data in this paper also prove this negative relationship between FOI and FOS.

Finally, the mediation effect test process and results once again confirm the above analysis. From the perspective of correlation, there is a negative correlation between FOI and sustainability. But in practice, this negative correlation is not significant after excluding the effects of FOS. The reason is that the negative impact of FOI on sustainability is mainly due to its resource competition with FOS. In the consumer industry under the new crown pneumonia epidemic and Demand-Side Reform, enterprises need to use FOS to ensure their basic viability. Especially for small and medium-sized enterprises, only by getting through the survival difficulties first can they better seek in-novation and development. These findings not only theoretically enrich the relation-ship between FOI, FOS, and sustainability, but also have positive implications for the COVID-19 response in other industries.

5.2. The Impacts of Firm Value

Maximizing firm value is the main goal of financial management and an important basis for the board of directors to evaluate managers [87-89]. In this study, we use the method of Wind database company who calculated firm value from the sum of the value of equity and the value of liabilities. According to this method, companies with higher enterprise value are generally stronger in size and profitability, and tend to have stronger viability and sustainability. The descriptive statistical results in this paper also confirm the judgement that corporate value and sustainability have a significant positive correlation. In fact, the statistics of our study also manifest that corporate value by itself does not have a significant impact on sustainability. The reason for the positive relationship between it and sustainability is, on the one hand, that it is closely related to factors such as the asset-liability ratio, the total number of employees, the proportion of executives, the proportion of independent directors, etc., all of which have a significant impact on corporate sustainability. On the other hand, in the context of Demand-Side Reform, corporate value may moderate the positive effect of FOS on sustainability.

For firms of consumer goods industry in COVID-19, Demand-Side Reform is a creative temporary subsidy mechanism undertaken by the Chinese government. Prior to the implementation of demand-side reform, the consumer goods industry had al-ready been severely affected by the COVID-19 pandemic. Seizing the Demand-Side Reform opportunities provided by the government and making in-depth efforts in marketing and promotion is an important way for enterprises to remove inventory and increase sales revenue from their main business. In this regard, there are significant differences in the impact of FOS strategies on sustainability between companies with large market value and those with small market value. The statistical analysis results of this paper using consumer goods industry data show that the market value of firms would positively moderate the relationship between FOS and sustainability. That is, the higher the market value of a business, the stronger the impact of FOS on sustainability.

The reason is that the value of the enterprise is not only the embodiment of the value of the enterprise's own assets and liabilities, but also reflects the true views of the public and the market on the capital strength and brand influence of the enterprise. Compared with enterprises with low market value, enterprises with high market value have more advantages in terms of customer recognition, social public relations and market influence, and the FOS strategy can play a better role. Many news reports have also shown that in the new crown pneumonia epidemic, large and medium-sized enterprises with high market value are the main participants in Demand-Side Reform, and together with government departments, they have provided many special subsidies to consumers and achieved a rebound in product sales revenue [90-92]. In contrast, small and medium-sized enterprises, due to limited resources, capabilities, and weak social network, even if they have been tilted in terms of FOS strategy, still have disadvantages over large enterprises.

In the statistical analysis of this paper, we also find that the moderating effect of firm value on the relationship between FOI and sustainability is insignificant. Combined with the above conclusion that the FOI strategy does not have a significant im-pact on sustainability after controlling FOS and other variables, it shows that neither companies with high market value nor those with low market value may significantly influence sustainability through FOI. In the context of covid-19 and demand-side re-form, large enterprises with high market value and small and medium-sized enterprises with low market value are facing a certain degree of resource shortage. There-fore, when many favorable opportunities suddenly arise in the market environment, by shifting resources and strategies from FOI to FOS, both the two kinds of firms may re-duce the negative correlation between FOI and sustainability and promote their sustainability. However, it should be noted that this strategic shift is not conducive to corporate innovation and high-quality development of the consumer goods industry in the long run. In China, the transformation and upgrading of the consumer goods industry is the trend of the times. After COVID-19, resource constraints would be eased as the economy improves, and companies should insist on using innovation as the main way to grow profits, or at least maintain a modest investment in innovation to provide better products and services.

Finally, in the Conclusions section, we have rearranged the conclusions based on your comments and that of another reviewer. The revised version is as follows:

  1. Conclusions

In the context of Demand-Side Reform, opportunities from the market side have increased, and sales growth has become more attractive to enterprises. On this back-ground, should companies focus on sales or innovation and how would they impact sustainability? This paper analyzes the strategic choices of enterprises and the moderating effect of corporate value, and obtains three main conclusions. First, in the context of Demand-Side Reform, for the consumer goods industry, the impact of FOI on corporate sustainability is negative, and the impact of FOS on corporate sustainability is positive. From the perspective of improving the sustainable development ability of enterprises, it is more appropriate for enterprises to choose FOS than FOI. Second, FOI does not have a direct negative impact on the sustainability of the business. The reason why it has a significant negative correlation with sustainability is mainly because it has a significant negative impact on FOS, which indirectly inhibits the sustainability of the enterprise. For the consumer goods industry under the Demand-Side Reform, be-cause the opportunities in the market are short and huge, and the changes in innovation are slow and too late to respond, the sustainability of the enterprise can be pro-moted by focusing on sales. Third, although corporate value does not significantly moderate the relationship between FOI and sustainability, it has a significant positive moderating effect on the positive relationship between FOS and sustainability. In the context of Demand-Side Reform, large enterprises can obtain greater sustainability improvements than small enterprises through FOS.

Most of the existing research focuses on environmental sustainability issues, while ignoring sustainability at the social and corporate governance levels. This has created confusion among consumers and businesses about how to manage sustainability [93-95]. Especially for the consumer goods industry, which faces relatively fewer environmental regulations than industries with more prominent environmental problems such as heavily polluted industries, sustainability at the social and corporate governance levels is more important. The relationship theory of environmental regulation and innovation and performance in the Porter hypothesis [96] has good convincing power in high-pollution and energy-consuming industries, but it does not fully ap-ply to consumer industries under the COVID-19 and the Demand-Side Reform as there are fewer environmental regulations. The research in this paper confirms the role of FOS in promoting sustainability, which is consistent with Melander and Pazirandeh [43] and Ferrer et al. [38] 's conclusions about the impact of sales and marketing relationships on sustainability. However, the research in this paper also shows that the impact of FOI on sustainability is not significant under the influence of limited re-sources and short-term opportunities in the market, and even negative correlation will occur, mainly through FOS. This is a useful complement to the findings of studies such as Forcadell et al. [19] and Desiana et al. [97] that innovation in general contexts significantly contributes to sustainability.

There are two main limitations to this study. First, considering the availability of data, the sample companies selected in this study are all listed companies. However, for the consumer goods industry, there are more small unlisted businesses. This paper fails to study how these companies should respond to the market opportunities brought about by Demand-Side Reform, and it is not known whether the research conclusions apply to these enterprises. Second, under the influence of the new crown pneumonia epidemic in 2019, all walks of life are generally facing the problem of declining demand and insufficient consumption. When the government makes new policies and boosts consumption, should companies continue to keep investing in innovation or focus on market sales? Obviously, the consumer goods industry is most directly affected by the government's Demand-Side Reform policies, which is the main reason why the consumer goods industry was selected as a sample in this study. But for other industries, are there similar conclusions to this paper in terms of FOI and FOS corporate decisions? This remains to be explored. We hope that subsequent studies will be able to explore in depth in both above areas. More in-depth and more general, if possible, we hope that follow-up research will be able to explore the relationship among market opportunity intensity, FOI and FOS decision-making, and corporate sustainability.

Finally, we would like to thank the reviewer for the careful reading and time again, again, and again!

Best Regards!

                                                      The Authors

Round 2

Reviewer 1 Report

The text is not properly ordered and is not adapted to the previously formulated remarks:

1st part 4 is titled: Results and Discussion - there are only results, no discussion.

2nd part 5 is entitled: Discussions Conclusions - there is only competition.

3rd part 6 is Conclusions - but there is a discussion.

4th the part about limiting inference is still missing.

Author Response

Dear Reviewer,

       Thanks for your time in reviewing again. However, we are concerned that because we used the Track Changes mode of Mirosoft Word, you are looking at the wrong version form.

       We are sorry for the confusion. The final version of this revision can be viewed in the following order of buttons in Microsoft Word after opening the file: Review--No Markup (near to Track Changes). You can also view the PDF version directly from the attachment.

      The reason we have this question is that your comments are inconsistent with the content of our revised manuscript. Specifically, the title of part 4 is "Results", the title of part 5 is "Discussions", and the title of part 6 is "Conclusions". If you confirm that you are reading the correct version, we will then modify it again.

Best Regards!

The Authors.

Reviewer 3 Report

Thanks for the revised version. Indeed, the overall quality of the paper is better.

Author Response

Thanks again and best regards!

Round 3

Reviewer 1 Report

The division of content between the Discussion and the Conclusion is still not transparent. In Conclusion, there is a discussion, and the Discussion lacks concrete results, recapitalization of research.

Maybe the wrong file has been re-attached?

Author Response

Dear Reviewer,

       How about integrating the discussion part in the "Conclusion" section into the "Discussion" section? In the first round of review, you suggested that there should be a discussion in the concluding part (we understand that because there is no separate discussion section in the paper), and another reviewer suggested adding a special section for discussion.

      In order to meet both of these requirements, we have added these two parts at the same time. Now, that reviewer (Reviewer 3) has approved our revision. But the two parts do seem to be somewhat repetitive.

     After we adjusted it in this way, the conclusion part contains only two parts: the summary of the whole text, and the research limitations and future directions. Do you think this is better?

Best Regards!

The Authors.

Round 4

Reviewer 1 Report

In the current text, the Discussion and Conclusion are properly separated. Thank you for your reply to the review.